# Investigation on Mechanism of Microstructure Evolution during Multi-Process Hot Forming of GH4169 Superalloy Forging

**DOI:** 10.3390/ma17071697

**Published:** 2024-04-07

**Authors:** Ming-Song Chen, Hong-Wei Cai, Yong-Cheng Lin, Guan-Qiang Wang, Hong-Bin Li, An Liu, Ze-Hao Li, Shan Peng

**Affiliations:** 1Light Alloy Research Institute, Central South University, Changsha 410083, China; 213812032@csu.edu.cn (H.-W.C.); 233811004@csu.edu.cn (A.L.); zehaolee@csu.edu.cn (Z.-H.L.); 223812057@csu.edu.cn (S.P.); 2School of Mechanical and Electrical Engineering, Central South University, Changsha 410083, China; 3State Key Laboratory of Precision Manufacturing for Extreme Service Performance, Changsha 410083, China; 4College of Metallurgies and Energy, North China Science and Technologies University, Tangshan 063009, China; yjsysz@ncst.edu.cn

**Keywords:** GH4169 superalloy, multi-process hot forming, dynamic recrystallization nucleation, δ phase

## Abstract

Typically, in the manufacturing of GH4169 superalloy forgings, the multi-process hot forming that consists of pre-deformation, heat treatment and final deformation is required. This study focuses on the microstructural evolution throughout hot working processes. Considering that δ phase can promote nucleation and limit the growth of grains, a process route was designed, including pre-deformation, aging treatment (AT) to precipitate sufficient δ phases, high temperature holding (HTH) to uniformly heat the forging, and final deformation. The results show that the uneven strain distribution after pre-deformation has a significant impact on the subsequent refinement of the grain microstructure due to the complex coupling relationship between the evolution of the δ phase and recrystallization behavior. After the final deformation, the fine-grain microstructure with short rod-like δ phases as boundaries is easy to form in the region with a large strain of the pre-forging. However, necklace-like mixed grain microstructure is formed in the region with a small strain of the pre-forging. In addition, when the microstructure before final deformation consists of mixed grains, dynamic recrystallization (DRX) nucleation behavior preferentially depends on kernel average misorientation (KAM) values. A large KAM can promote the formation of DRX nuclei. When the KAM values are close, a smaller average grain size of mixed-grain microstructure is more conductive to promote the DRX nucleation. Finally, the interaction mechanisms between δ phase and DRX nucleation are revealed.

## 1. Introduction

GH4169 alloy is a typical Ni-Fe-Cr-based superalloy, which can maintain excellent comprehensive mechanical properties and microstructure stability under high temperatures and high pressure [1,2,3,4]. Hence, GH4169 superalloy is commonly utilized in the production of critical components in the energy and aerospace industries, such as casings and turbine discs [5,6,7,8]. In general, hot deformation, such as die forging and rolling, is the main way to manufacture these components [9,10,11]. Therefore, the flow behavior of the GH4169 superalloy during hot deformation has drawn the attention of several academics. Jarugula et al. [12] and Voyiadjis et al. [13] developed phenomenological and physics-based constitutive models, respectively. Wang et al. [14] considered the effect of δ phase (Ni_3_Nb) when establishing the constitutive model. Wen et al. [15] optimized the thermal deforming parameters and the aging time of different processes through the thermal processing maps.

In addition, dynamic recrystallization is considered as an important microstructure evolution mechanism of nickel-based alloys during thermal deformation [16,17,18,19]. The high DRX degree is beneficial to the uniformity of the grain structure after thermal deformation [20,21]. Therefore, plenty of investigations on the DRX behavior in nickel-based alloys have been undertaken, such as Inconel X-750 [22], Alloy 690 [23], Alloy 617 [24], C276 [25], Incoloy 901 [26], XH55 [27] and Haynes 282 [28]. GH4169 superalloy is one of the most important superalloys in the aviation field. Its DRX behavior is also the focus of many scholars. Chen et al. [29] studied the influence of hot deformation parameters on the DRX of GH4169 superalloy and revealed the evolution mechanism of the dislocation substructure. The effect of deformation parameters on the the DRX mechanisms of Inconel 718 was analyzed by Azarbarmas et al. [30], and discontinuous dynamic recrystallization (DDRX) is considered as the main nucleation mechanism. Geng et al. [31] established a DRX kinetic model of the solution-treated GH4169 superalloy. Furthermore, the δ phase plays a significant role in the DRX behavior of GH4169 superalloy. It can promote DRX nucleation, but hinders the growth of DRX grains [32,33,34]. Zhang et al. [35] delved into the influence of δ phases on DRX and improved the cellular automata (CA) model of DRX. Wen et al. [36] developed a DRX kinetic model that considers the initial δ phase content. Páramo-Kañetas [37] found that the δ phases with long needle shape inside the grain are beneficial to obtaining an even grain microstructure after thermal deformation. Typically, the manufacturing of most GH4169 superalloy parts often includes multiple thermal deformation processes [38,39]. During multi-process hot forming, the microstructure evolution is significantly complex. For example, the uneven strain distribution of the pre-deformed forging will affect the microstructure evolution of subsequent processes, especially the DRX behavior during the final deformation. However, existing studies only concentrate on the DRX behavior of GH4169 superalloy during a single hot deformation. Hence, it is essential to reveal the mechanism of microstructural evolution during multi-process hot forming of GH4169 superalloy and the influence of different microstructures, formed at each stage, on DRX behavior during the final deformation.

In this study, firstly, the experimental billet was subjected to hot upsetting (pre-deformation). Subsequently, aging treatment (AT) to precipitate enough δ phases, high temperature holding (HTH) to uniformly heat the forging, and hot compression (final deformation) experiments were done on small specimens that were cut from different deformation zones of the large forging. The microstructure evolution throughout the hot working processes was studied. In addition, the effect of different grain microstructures on DRX nucleation during final deformation was investigated. Ultimately, this study elucidated interaction mechanisms between the δ phase and DRX nucleation during hot deformation.

## 2. Materials and Experiments

Table 1 lists the detailed chemical compositions of a commercial GH4169 superalloy that is provided by Fushun Special Steel Co., Ltd. in Fushun of China. used in this investigation. From the original bar, an experimental billet measuring ϕ 100 mm × 140 mm was cut by a CNC wire cutting machine. Then, the experimental billet was subjected to solution treatment. In the solution treatment, the billet is held at 1040 °C for 85 min. Typically, annealing twins are observed within equiaxed grains and have mutually parallel grain boundaries. As displayed in Figure 1, the grain microstructure of GH4169 superalloy after solution treatment consists of equiaxed grains and annealing twins. In addition, the δ phases were almost completely eliminated. Subsequently, hot upsetting (i.e., pre-deformation) was applied to the experimental billet, reducing its height to 70 mm. The temperature of hot upsetting was 950 °C and the strain rate was 0.1 s^−1^. Figure 2a displays the forging after hot upsetting. The strain distribution of the forging is shown in Figure 2b. According to the value of the initial strain, three deformation zones are divided. The areas marked by the black dotted line and the red dotted line are the small deformation zone and the large deformation zone, respectively, and the rest is the middle deformation zone. Several cylindrical samples measuring ϕ 10 mm × 15 mm were cut from the three deformation zones of the forging. The samples cut from the small deformation zone, medium deformation zone and large deformation zone were defined as Sample I, Sample II and Sample III, respectively. To ensure similarity in the microstructures of samples from the same deformation zone, the samples were cut along the same circumference and height. Then, all samples were firstly held at 720 °C for 8 h, so that γ″ phases could be uniformly precipitated in the microstructure. After the temperature was raised to 900 °C, all samples were held for another 9 h to make γ″ phases transform into δ phases. Using this aging treatment (AT) method can make the distribution of δ phase precipitated in the microstructure more uniform. After AT, the hot compression experiments with different deformation parameters were done by using a Gleeble-3500 (Poestenkill, NY, USA). The detailed experimental procedure after pre-deformation is described in Figure 3, where WQ in Figure 3 refers to water quenching.

In this study, to obtain the microstructures of samples, we used optical microscopy (OM), Scanning Electron Microscopy (SEM), Electron Backscattered Diffraction (EBSD), and Transmission Electron Microscopy (TEM). For SEM and OM, the samples were firstly mirror polished, and then the samples were electrolytically corroded in an etching solution composed of 70 mL H_3_PO_4_ and 30 mm H_2_O. The voltage of electrolytic corrosion was 3 V and the electrolytic corrosion time was about 10–25 s. In the TESCAN MIRA4 LMH Scanning Electron Microscope (TESCAN, Brno, Czechia), several images with magnifications of 1000×, 2000× and 5000× were acquired using secondary electron signals. The model of the OM device was a Keyence VHX-5000 (KEYENCE, Osaka, Japan). For EBSD and TEM, discs with a diameter of 3 mm were cut from smooth flakes with a thickness of 70–80 mm with a hole punch, and then the discs were electrolytically polished by a solution of 10 mL HClO_4_ and 90 mL CH_3_CH_2_OH. The temperature of electrolytic polishing was −40 to −25 °C and the voltage was 25 V. The model of the EBSD equipment was a NordlysMax2 (Oxford Instruments, Oxford, UK). In EBSD characterization, the observation area was a square of 250 μm × 250 μm and the scanning step was set as 1 μm. All EBSD data were analyzed on the MTEX-5.7.0 toolbox. When reconstructing grains, low-angle grain boundaries (LAGBs, 2° ≤ θ < 15°) are represented by gray lines and high-angle grain boundaries (HAGBs, θ > 15°) are represented by black lines. The model of the TEM device was Talos FEI 200X (Thermo Fisher Scientific, Waltham, MA, USA).

## 3. Experimental Results

### 3.1. Microstructure Evolution during Pre-Deformation

The microstructure of different regions of the upsetting forging is shown in Figure 4. As for sample I, bulged grain boundaries can be observed in Figure 4a. Due to the small initial strain, there is no obvious dynamic recrystallization behavior. Consequently, the microstructure of sample I primarily consists of equiaxed grains and annealing twins. However, some elongated grains are formed in samples II and III (as displayed in Figure 4c,e). At the same time, many DRX grains are produced with the stimulation of deformation and thermal energies, which is the typical forged mixed-grain microstructure. Because of the larger initial strain, more deformation energy are produced in sample III. Therefore, the DRX behavior is stronger. Relatively, more DRX grains can be observed in the microstructure of sample III.

In addition, from Figure 4b,d,f, it can be observed that some δ phases come out at grain boundaries in the microstructure of these samples. This is because a small activation energy is required for δ phase to nucleate at grain boundaries [40]. Thus, δ phase nuclei are more likely to form at grain boundaries.

### 3.2. Microstructure Evolution during AT

In this section, we aim to study the evolution of δ phase in different deformation regions. Figure 5 displays the distribution and the δ phase content of samples in different deformation regions after AT. As displayed in Figure 5a–c, after AT, there are δ phases densely distributed in three kinds of samples. The contents of δ phase in samples I–III are 29.73%, 27.38% and 26.02%, respectively, as illustrated in Figure 5d. Moreover, intragranular δ phases are mainly distributed with a needle-like shape, while the δ phases on grain boundaries are distributed with a short rod-like shape. This phenomenon is more obvious when the initial strain is larger. This is because the deformation energy promotes the growth of DRX grains during AT, which weakens the precipitation behavior of δ phase.

Comparing the distribution of δ phase in the microstructures of different deformation regions after AT, it can be found that the δ phases in the microstructure of sample I are more evenly distributed. The reasons for this can be summarized as follows. In small deformation region, the content of defects, such as dislocations and vacancies, is less. As a result, for the sample obtained from the small deformation region, the evolution of the second phase is dominated by the transformation of γ″ phases to δ phases when the aging temperature is increased to 900 °C. Yet, there are more dislocation and vacancies in samples II and III. In these crystal defects, many δ phase nuclei can be induced to form. Thus, the δ phases in the region with plentiful defects are densely distributed. This results in relatively inhomogeneous distribution of δ phase in samples II and III.

### 3.3. Microstructure Evolution during HTH before Final Deformation

#### 3.3.1. Effect of Initial Strain

Figure 6 illustrates the microstructures of aged samples in different deformation regions after HTH, where the parameters of HTH are “980 °C × 5 min”. From Figure 6a, it can be found that there are few recrystallized grains in the microstructure of sample I. Furthermore, after holding at 980 °C for 5 min, the recrystallized grains grow slightly in samples II and III (as shown in Figure 6d,g). Because of the different recrystallization degrees, the average grain sizes of samples I–III are significantly different, and can be ascertained as 41.44 μm, 29.09 μm and 20.62 μm, respectively. Notably, there is also a significant difference in the number of newborn grains, whose size is defined as 1–5 μm. When the initial strain is augmented, the number of newborn grains increases greatly. This is attributed to more deformation energy being generated during thermal deformation, which promotes the nucleation of SRX during HTH.

Furthermore, the dissolution behavior of δ phase in the sample III is greatly different from those of samples I and II. Seen from Figure 6i, obvious dissolution behavior of the δ phase occurs in sample III after holding at 980 °C for 5 min. However, for samples I and II, a small quantity of δ phases is dissolved (Figure 6c,f). There are two reasons for this phenomenon. Firstly, compared with samples I and II, there are more δ phases with short rod-like shape in sample III after AT (as demonstrated in Figure 5a–c). Compared to the short rod-like δ phase, the dissolution rate of long needle-like δ phase occurs more slowly. Secondly, the difference in dislocation content is also a crucial reason. Because the initial strain of sample III is larger than those of samples I and II, there are more dislocations in sample III after pre-deformation, which accelerates the diffusion of Nb atoms and then promotes the rapid dissolution of δ phase [41].

As presented in Figure 6b,e,h, after HTH, the large kernel average misorientation (KAM) values of samples I–III can be calculated as 0.88°, 1.25° and 1.24°, respectively. Due to the smallest initial strain, the KAM of sample I is the smallest. Although the initial strain of sample III is larger than that of sample II, the KAM values of samples II and III are very similar. The reason is that the recrystallization behavior and the dissolution behavior of δ phase in sample III are more significant during HTH. This leads to a significant consumption of deformation energy.

#### 3.3.2. Effect of Holding Time

Figure 7 illustrates the grain microstructure of aged sample II after holding at 980 °C for different times. As displayed in Figure 6e and Figure 7b,e,h, with the increase in the holding time, the recrystallized grains gradually grow, and the recrystallized grains are mainly distributed on the boundaries of deformed grains. Additionally, the change in the average grain size shows a decreasing trend when the holding time is extended (as shown in Figure 6a,d,g). However, there are a few large deformed grains in the microstructure, although the holding time reaches 60 min, which means that full recrystallization is not achieved.

From Figure 6f and Figure 7c, δ phases around the grain boundaries are preferentially dissolved when the holding time is augmented from 5 min to 15 min. The reason is that there are mostly rod-like and spherical δ phases near the grain boundaries, and their dissolution time is shorter than that of long needle-like δ phases [42]. When the holding time reaches 30 min, the δ phases near the deformed grain boundaries are almost entirely dissolved, and there are obvious signs of dissolution of intragranular δ phases (as illustrated in Figure 7f). In addition, as the holding time is further prolonged to 60 min, the content and shape of the δ phase changes slightly (as demonstrated in Figure 7i). This is because the δ phase remains in a dynamic equilibrium state when the aging pretreated GH4169 superalloy is held at 980 °C for more than 30 min [43].

Moreover, to research the effect of holding time on the evolution of microstructure, the changes in KAM value and recrystallization fraction at different holding times at 980 °C are analyzed (as illustrated in Figure 8). At the initial stage of HTH (*t*_hold_ ≤ 15 min), recrystallization behavior and the dissolution of δ phase leads to much deformation energy being consumed. Then, when the holding time is 30 min, the KAM value reduces to 0.69, which indicates that the content of residual deformation energy is very low. Further prolonging the time, the behavior of static recrystallization nucleation is weakened due to the dissolution of δ phase and the consumption of deformation energy. The trend of the number of newborn grains can confirm this idea. Meanwhile, the reduction in deformation energy leads to the migration rate of grain boundary being slowed down. Thus, the microstructure evolution is dominated by the slow growth of recrystallized grains, and the recrystallization fraction only increases by 10.1%.

### 3.4. Microstructure Evolution during Final Deformation

#### 3.4.1. Effect of Initial Strain

In order to study the effect of initial strain on the evolution of microstructure during final deformation, the specimens after HTH were subjected to thermal compression (i.e., final deformation), where the parameters of HTH were “980 °C × 5 min” and deformation parameters were “*T* = 980 °C, $\dot{\varepsilon }=0.1$ s^−1^, *ε* = 0.69”.

Figure 9 demonstrates the grain microstructures of samples I–III after final deformation. From Figure 9a,c,e, there are many low-angle grain boundaries (LAGBs) in the deformed grains, which indicates that there are abundant substructures (dislocation networks and sub-grains, etc.) in the deformed grains. As demonstrated in Figure 9a,c, in the grain microstructures of samples I and II, obvious bulged grain boundaries (marked by the orange arrows) can be observed, and most DRX grains are formed on the boundaries of deformed grains. These phenomena belong to typical characteristics of DDRX [44].

Typically, the higher KAM value means the higher content of residual deformation energy in the microstructure [45]. The KAM value of sample III is significantly lower than that of samples I and II, as shown in Figure 9b,d,f. Meanwhile, the number of fine grains in sample III is significantly more than those of samples I and II. This indicates the more violent DRX nucleation behavior of sample III during the final deformation compared with that of samples I and II, which is required to consume much deformation energy.

As demonstrated in Figure 9a,c,e, the average grain sizes of samples I–III after final deformation are 24.48 μm, 7.22 μm and 3.51 μm, respectively. The average grain size of sample I is the largest because the content of small recrystallized grains in its microstructure is the least. However, the average grain size of sample III after deformation is only 3.51 μm (Figure 9e). This is because there are plenty of fine DRX grains in the microstructure of sample III. Moreover, according to its grain size distribution diagram, it can be found that the size of most of the grains in sample III is concentrated in the range of 1–5 μm. Hence, after final deformation, the average grain size of sample III is the smallest.

#### 3.4.2. Effect of Final Strain

In order to study the influence of the final deformation strain on the microstructure of the same deformation region of the pre-deformed forging, several samples II were subjected to final deformation with different true strains (i.e., final strain).

The grain microstructures of the samples II after final deformation with different final strains are shown in Figure 10, where the deformation temperature and strain rate are 980 °C and 0.1 s^−1^, and the parameters of HTH are “980 °C × 5 min”. From the red frame in Figure 10b, it can be observed that several deformed grains with small size are distributed near the boundaries of large deformed grains. This is because the recrystallized grains that grew up during HTH become small deformed grains due to the progress of final deformation. As illustrated in Figure 10a, DRX grains are almost always formed on the boundaries of deformed grains and almost not formed within deformed grains when the final strain is 0.36. However, as the final strain is raised to 0.69, a few of the DRX grains can be clearly observed inside the deformed grains. In addition, it can be found that small DRX grains begin to aggregate together. In the deformed sample with a final strain of 1.20, this phenomenon becomes more obvious. The microstructure consists of mostly fine recrystallized grains and a small part of deformed grains (as displayed in Figure 10d). Correspondingly, the KAM value is reduced to 0.69. It is worth noting that there are some unclosed HAGBs in the deformed grains of samples with different final strains. According to the research of Kaibyshev et al. [46], LAGBs and MAGBs inside deformed grains will become unclosed HAGBs through continuously absorbing dislocations, which is the phenomenon caused by CDRX behavior. Furthermore, in the samples with final strains of 0.69 and 1.20, there are some small DRX grains at the unclosed HAGBs. These phenomena suggest that CDRX behavior can provide new nucleation sites for DDRX grains [47].

As can be seen from Figure 9c and Figure 10a,c, the average grain sizes are calculated as 12.48 μm, 7.22 μm and 4.32 μm, with the augment of final strain. This is attributed to the increase in final strain, resulting in more deformation energy generated, which significantly promotes the formation of fine DRX grains in the microstructure. For samples with final strains of 0.36 and 0.69, uneven grain microstructures are obtained due to the low degree of recrystallization. However, in the sample with a final strain of 1.20, the grain size is concentrated in the range of 1–10 μm, and the uniformity of the grain microstructure is the best.

In order to further discuss the DRX behavior during the final thermal deformation, it is necessary to analyze the distribution of the misorientation angle. As shown in Figure 11, with the rise in final strain, the frequency of LAGBs shows a decreasing trend, while the frequency of HAGBs shows an increasing trend. The average misorientation angles ($\bar{\theta }$) are 16.28°, 23.84° and 36.77°, respectively. Typically, the medium angle boundary (MAGB), with grain boundary misorientation angles of 10–15°, is the sign that continuous dynamic recrystallization (CDRX) behavior is occurring [48]. Compared with the sample deformed at a final strain of 0.36, there are more MAGBs in the sample with a final strain of 0.69. This is because the continuous dislocation proliferation promotes the rotation of sub-grains, forming DRX grains. Namely, the high strain can promote CDRX behavior, whereas, when the final strain is 1.20, the frequency of MAGBs drops slightly. It could be that some sub-grains have completed the transformation into DRX grains. In addition, the frequency of MABGs after final deformation fails to exceed 5%. Therefore, it can be inferred that the nucleation mechanism of DRX during the final hot deformation of GH4169 superalloy is dominated by DDRX, and CDRX behavior can assist the nucleation of DDRX grains.

Figure 12 shows the SEM micrographs of the samples II after final deformation with different final strains. As shown in Figure 12, after the final deformation, some δ phases with long needle shape fracture into δ phases with short rod shape. Moreover, the content of δ phase decreases slightly along with the rise in the final strain. This is because larger final strain means longer deformation time, which can provide a longer time for the dissolution of δ phase. In the sample with a final strain of 1.20, there are plenty of δ phases with short rod shape. And the long needle-like δ phases undergo obvious distortion and folding at the large strain. Nevertheless, although the final strain is augmented to 1.20, the distribution of δ phase is still very dense. Due to the strong pinning effect of closed-set δ phases, the growth of these recrystallized grains is strongly inhibited [49]. Therefore, even though the final strain increases, only slight growth of recrystallized grains occurs.

#### 3.4.3. Effect of Holding Time

The effect of the holding time before deformation on the deformed grain microstructure is displayed in Figure 13, where the holding temperature is 980 °C and deformation parameters are “*T* = 980 °C, $\dot{\varepsilon }=0.1$ s^−1^, *ε* = 0.69”.

From Figure 13b,d,f, it can be found that the quantity of small-sized deformed grains in the microstructure increases significantly along with the holding time increasing from 15 min to 30 min or 60 min. This is attributed to the grain microstructure being uniformly refined under the influence of SRX behavior with the longer holding time. Moreover, when the holding times of the samples II are 5 min, 15 min, 30 min and 60 min, respectively, the KAM values of deformed microstructures are 1.40°, 1.60°, 1.37° and 1.29°, respectively. Figure 14 demonstrates the effect of holding time on the DRX fraction of the deformed microstructure. The trend of the DRX fraction is close to that of the KAM value. When the holding time is augmented from 5 min to 15 min, the DRX fraction decreases slightly. Nonetheless, as the holding time is further augmented to 60 min, the DRX fraction greatly rises. As the holding time is prolonged, the δ phases gradually dissolve, and the refinement of the grain microstructure leads to an increase in the grain boundary area. During the final deformation, the larger grain boundary area provides more nucleation sites for DDRX grains [50], which promotes DRX behavior. Therefore, after final deformation, compared to other samples, the sample whose holding time is 60 min has a significantly higher DRX fraction.

## 4. Discussion

### 4.1. Effect of Different Grain Microstructures on DRX Nucleation during Final Deformation

After HTH, there are three typical grain microstructures: uniform coarse grain microstructure, mixed grain microstructure, and fine grain microstructure. After sample I was held at 980 °C for 5 min, its grain microstructure was coarse and uniform (Type 1). Different mixed grain microstructures are formed in the following situations: sample II is held at 980 °C for 5 min and 15 min (Type 2–3); sample III is held at 980 °C for 5 min (Type 4). After sample II was held at 980 °C for 60 min, its grain microstructure is fine (Type 5). Table 2 presents characteristics and corresponding processes for different types of grain microstructure after HTH.

To investigate the influence of different grain microstructures on DRX nucleation during the final deformation, the area frequency of new-born grains in the microstructures after final deformation is counted, as displayed in Figure 15. From Figure 15, it can be found that, after the final deformation, the area frequency of new-born grains of Type 1 is only 41.6%, which is significantly lower than that of the other types. The reason is that Type 1 has a low KAM value and a large average grain size after HTH (as displayed in Table 2). On the one hand, after HTH, the KAM value of Type 1 is low, which means there are few dislocations in the microstructure. Thus, compared with the other types, it is more difficult for Type 1 to reach the critical dislocation density for DRX nucleation. On the other hand, the small grain boundary area of Type 1 results in fewer nucleation sites for DDRX grains than the other Types. Consequently, the DRX nucleation behavior of Type 1 is the weakest during the final deformation. For Type 5, although its KAM value is the lowest, its grain boundary area is the largest. Therefore, the DRX nucleation behavior of Type 5 is strong. After final deformation, the area frequency of new-born grains reaches 77.4%.

When the mixed-grain microstructure undergoes the final deformation, the DRX nucleation behavior is dominated by the KAM value and average grain size. As depicted in Table 2, compared with Type 3, Type 2 has a higher KAM value, but a slightly larger average grain size. After final deformation, the area frequency of new-born grains of Type 2 is obviously higher than that of Type 3 (as presented in Figure 15). This indicates that during the final deformation of the mixed grain microstructure, the DRX nucleation behavior preferentially depends on its KAM value. The reason for this phenomenon can be summarized as follows. The high KAM value of the mixed grain microstructure means that the frequency of the high-energy region in the microstructure is large. During the final deformation, DRX grains are easily formed in these high-energy regions. The formation of DRX grains results in an increase in the number of grain boundary triple junctions. During the thermal deformation of nickel-based alloys, the first layer of DDRX grains is formed at the bulged grain boundaries, and the next layer of DDRX grains is formed at the grain boundary triple junction [30]. Namely, the high KAM value of the mixed grain microstructure can promote the formation of the second layer of DDRX grains. Furthermore, for Type 2 and Type 4, their KAM values are close, but the average grain size of Type 4 is smaller. After final deformation, the area frequency of new-born grains of Type 4 is significantly higher than that of Type 2. This indicates that the DRX nucleation behavior is determined by the average grain size when the KAM values of mixed-grain microstructures are close. When the average grain size is smaller, the DRX nucleation behavior during the final deformation process is stronger.

### 4.2. Interaction Mechanisms between δ Phase and DRX Nucleation

In order to study the interaction mechanism between δ phase and DRX nucleation, the bright-field TEM figures of samples II deformed at 980 °C–0.1 s^−1^ to different final strains are analyzed. From Figure 16a, it can be observed that there are abundant tangled dislocations between the parallel distributed δ phases. Compared with hard and brittle δ phase, the γ matrix is more prone to deformation. Meanwhile, the migration of dislocation will be hindered by δ phase. Therefore, during the thermal deformation, many dislocations generated due to the deformation of γ matrix will be accumulated between δ phases. As shown in Figure 16b, when the final strain reaches 0.69, plentiful dislocation cells are formed around δ phases. The meeting of dislocation cells leads to the formation of localized high-energy regions (marked by the red dotted ellipse), which can trigger the DDRX behavior. In addition, some dislocation cells can transform into sub-grains by absorbing the dislocation. Therefore, in Figure 17b, some sub-grains can be observed between the parallel distributed δ phases. The dislocation density in the regions where these sub-grains are located is relatively high, which reduces the critical size for the transformation of sub-grains into CDRX grains [51]. Namely, δ phase promotes the CDRX behavior. When the final strain is further raised to 1.20, the content of dislocations and substructures is significantly reduced, and meanwhile, more DRX grains are formed (as displayed in Figure 16c). Furthermore, compared with the short rod-like or the spherical δ phase, the long needle-like δ phase has a stronger promotion effect on the DRX behavior. Increasing the final deformation reduces the critical nucleation dislocation density, eliminating part of the long needle-like δ phases. Therefore, coordinating the deformation temperature and the δ phase morphology is the key to obtaining a uniform and fine deformation grain microstructure [52,53].

In addition, from Figure 17a, it can also be found that the decreasing trend of δ phase content gradually slows down as the final strain increases. When the final deformation begins, many dislocations will be generated around δ phases. However, most regions fail to reach the critical dislocation density for DRX nucleation in the microstructure. At this time, with the effect of dislocations, the diffusion rate of Nb atoms will be accelerated and δ phases will break [54]. Therefore, when the final strain increases from 0 to 0.36, the dissolution rate of δ phase is very fast. With further increase in the final strain, many dislocations are accumulated, which makes it easy to reach the critical dislocation density for DRX nucleation. Once DRX grains are formed near the δ phases, a part of the dislocations near the δ phases will be consumed (as depicted in Figure 16c), which can slow down the diffusion rate of Nb atoms. Furthermore, the fracture of δ phase leads to the increase in phase boundary area, which implies that the fracture of δ phase is required to drive many dislocations [55]. Namely, the occurrence of DRX nucleation can protect the δ phase, slowing down the dissolution rate of the δ phase. Thus, although the final strain increases to 1.20, the δ phase is still dominated by long needle-like shapes. In addition, as shown in Figure 17b, during the final deformation, the content of dissolved δ phases in samples II and III is lower than that in sample I. This indicates that the protective effect obtained by δ phases during thermal deformation is stronger when the KAM value of the microstructure before deformation is larger. From Figure 18, it can be found that DRX grains are formed at the boundaries of short rod-like δ phases, and most dislocations around δ phases are consumed.

## 5. Conclusions

Different from existing studies that only analyze the microstructure evolution of GH4169 alloy during a single hot deformation, this study focuses on the microstructure evolution during multi-process hot forming of GH4169 superalloy and the evolution of the microstructure in each process. Particular attention is paid to the effect of different grain microstructures on DRX nucleation during final deformation. Finally, interaction mechanisms between δ phase and DRX nucleation are revealed. Several important conclusions can be drawn, as follows:

(1) In the region with a large initial strain of the pre-forging, the dense δ phases precipitated in AT dissolve faster during HTH, but dissolve slowly during the final deformation. This is because the higher dislocation content results in a faster dissolution rate of the δ phase. However, during the final deformation, it is easier to reach the critical dislocation density for DRX nucleation due to more dislocations remaining in the region with a larger initial strain, which leads to the microstructure evolution being dominated by DRX nucleation. Meanwhile, the formation of many DRX grains attached to δ phases can protect δ phases, which weakens the dissolution behavior of the δ phase. The above mechanisms lead to the fine grain microstructure with short rod-like δ phases as boundaries being easy to form in the region with a large strain of the pre-forging after final deformation.

(2) During final deformation, the DRX nucleation behavior that occurs in the fine-grain microstructure is stronger than that in the uniform and coarse-grain microstructure. As for mixed-grain microstructures, DRX nucleation behavior preferentially depends on KAM values of mixed-grain microstructures. A large KAM can promote the formation of DRX nuclei. When the KAM values are close, the DRX nucleation behavior is dominated by the average grain size. When the average grain size of the mixed-grain microstructure is smaller, the DRX nucleation behavior is stronger.

(3) δ phases can cause dislocations to accumulate around them during hot deformation. Hence, it is easy for local high-energy regions and many substructures to be formed, which can promote DRX nucleation. Meanwhile, numerous dislocations near δ phases are consumed due to the formation of DRX grains, which protects δ phases. When the KAM value of the microstructure before deformation is larger, the protective effect that δ phases obtain during thermal deformation is stronger.

## Figures and Tables

**Figure 1 materials-17-01697-f001:**
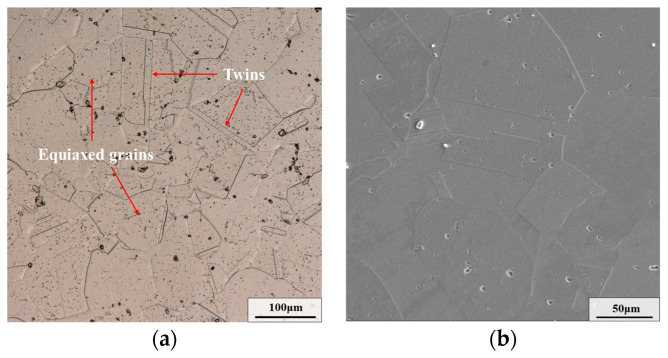
Microstructure of GH4169 alloy after solution treatment: (**a**) OM map; (**b**) SEM map.

**Figure 2 materials-17-01697-f002:**
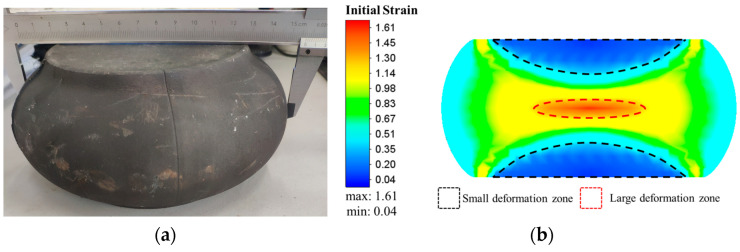
Maps of: (**a**) forging after hot upsetting; (**b**) strain distribution of the forging.

**Figure 3 materials-17-01697-f003:**
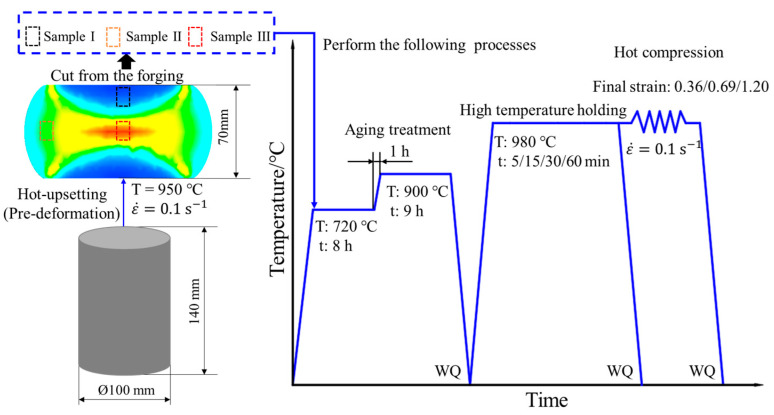
Detailed experimental procedure after pre-deformation.

**Figure 4 materials-17-01697-f004:**
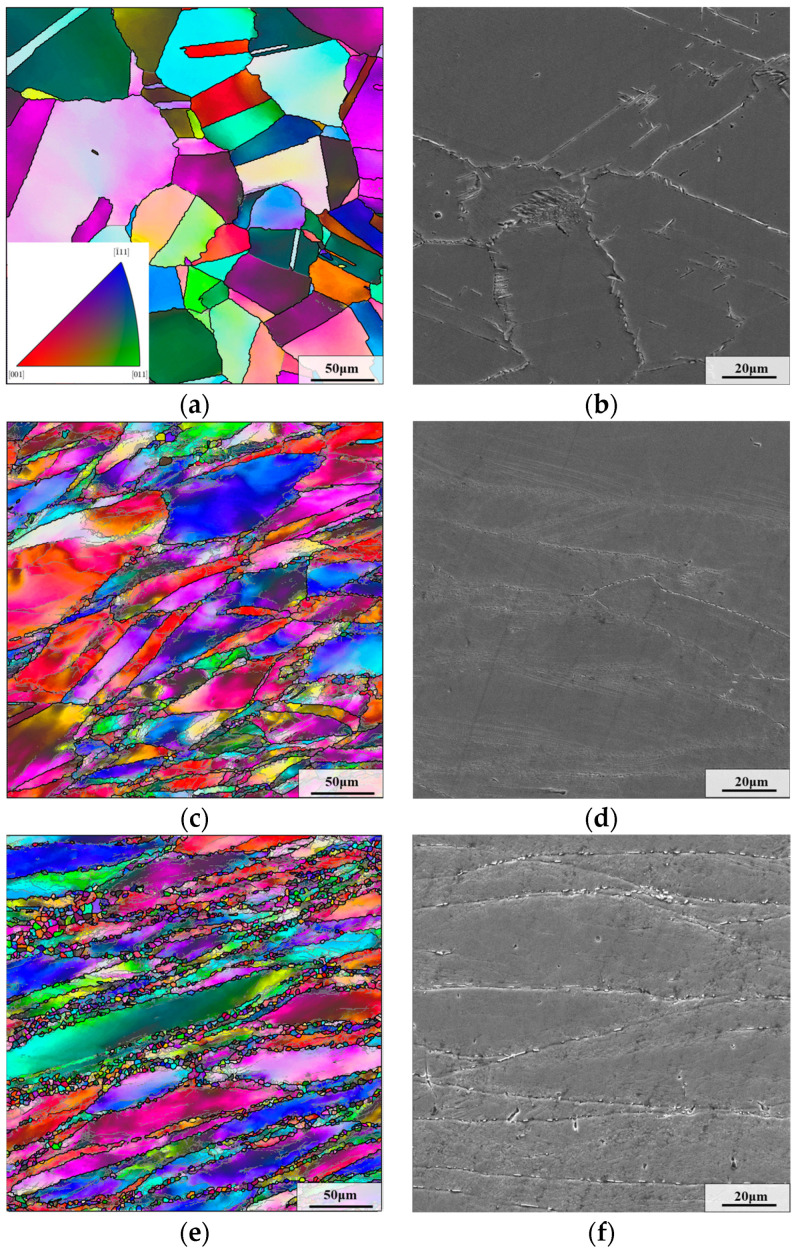
Microstructure of different regions after the first thermal deformation: (**a**,**b**) sample I; (**c**,**d**) sample II; (**e**,**f**) sample III.

**Figure 5 materials-17-01697-f005:**
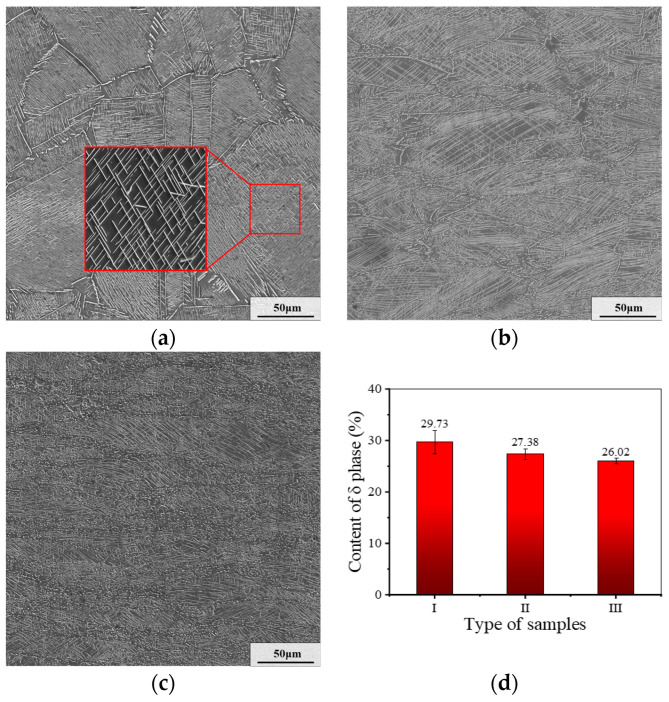
Distribution and content of δ phase for samples in different deformation regions after AT: (**a**) sample I; (**b**) sample II; (**c**) sample III; (**d**) content of δ phase. (The red box represents local magnification).

**Figure 6 materials-17-01697-f006:**
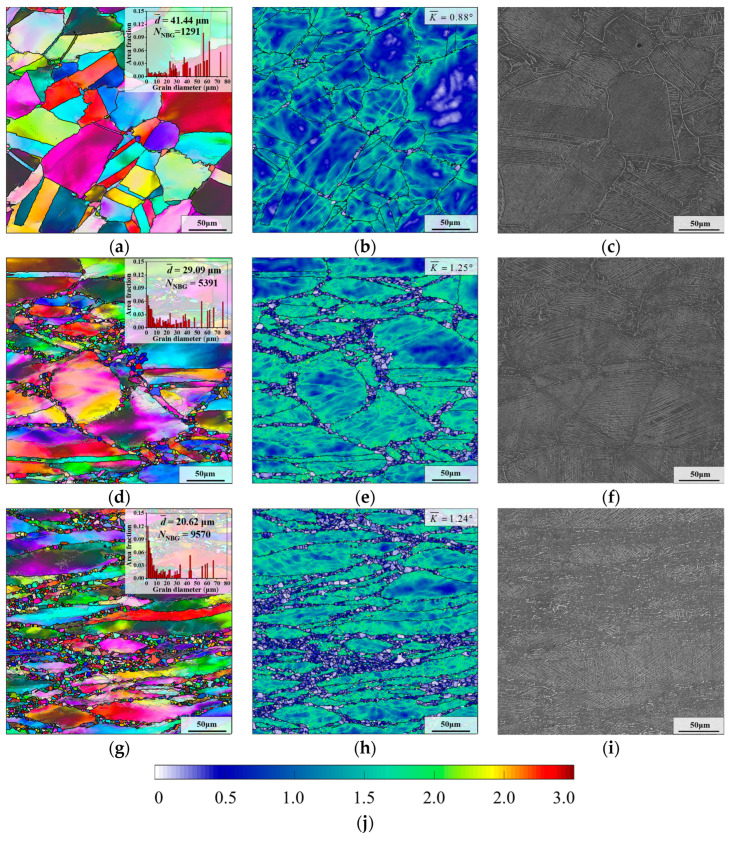
Microstructures of aged samples in different deformation regions after HTH: (**a**–**c**) sample I; (**d**–**f**) sample II; (**g**–**i**) sample III; (**j**) scale label of KAM maps.

**Figure 7 materials-17-01697-f007:**
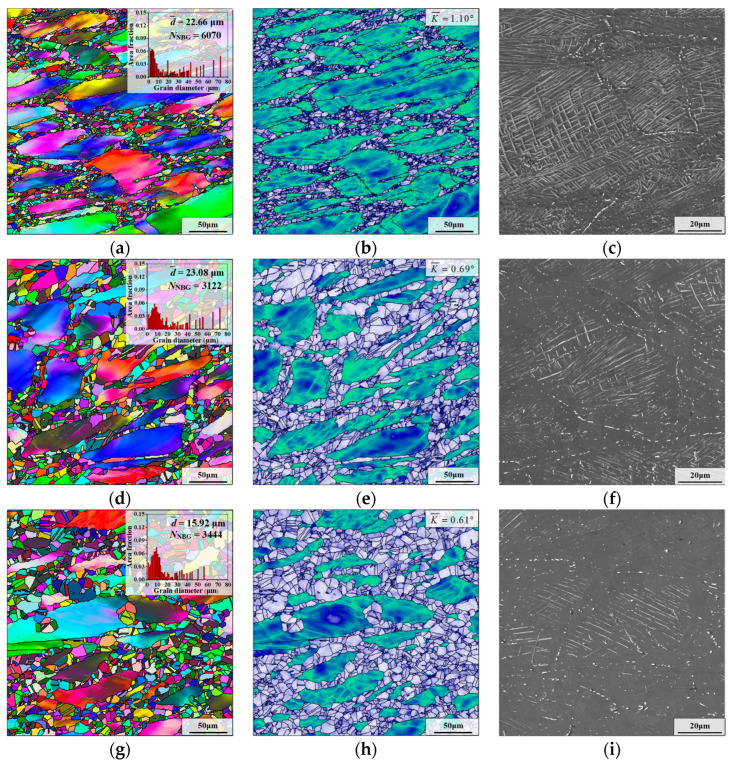
Microstructures of aged sample II after holding at 980 °C for different times: (**a**–**c**) *t*_hold_ = 15 min; (**d**–**f**) *t*_hold_ = 30 min; (**g**–**i**) *t*_hold_ = 60 min.

**Figure 8 materials-17-01697-f008:**
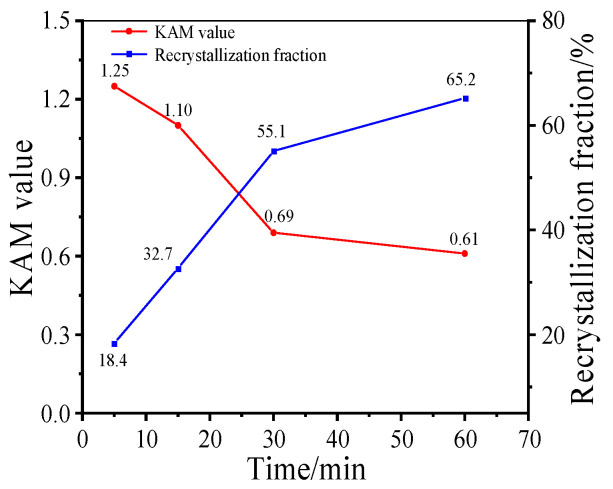
Curves of KAM value and recrystallization fraction of samples II after holding at 980 °C for different times.

**Figure 9 materials-17-01697-f009:**
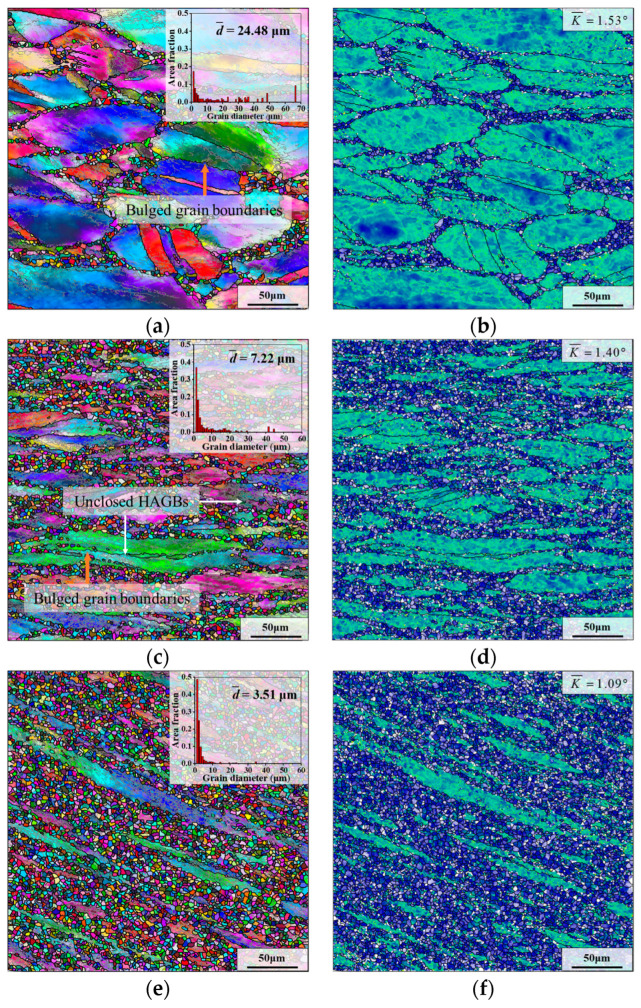
Grain microstructures of samples I–III after final deformation: (**a**,**b**) sample I; (**c**,**d**) sample II; (**e**,**f**) sample III. (The parameters of HTH are “980 °C × 5 min” and deformation parameters are “*T* = 980 °C, $\dot{\varepsilon }=0.1$ s^−1^, *ε* = 0.69”).

**Figure 10 materials-17-01697-f010:**
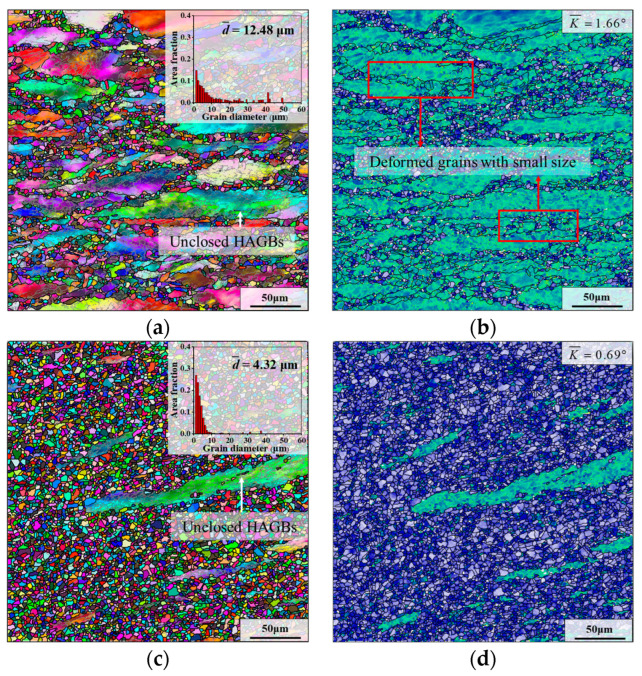
Grain microstructures of samples II after final deformation with different true strains (i.e., final strains) at 980 °C and 0.1 s^−1^: (**a**,**b**) *ε* = 0.36; (**c**,**d**) *ε* = 1.20. (The parameters of HTH are “980 °C × 5 min”).

**Figure 11 materials-17-01697-f011:**
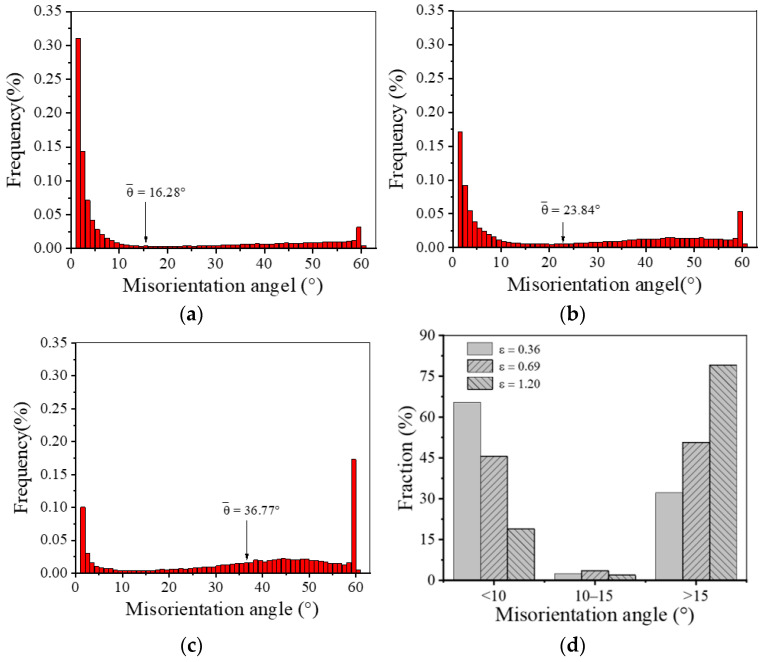
Misorientation angle distributions of the deformed samples with different final strains: (**a**) *ε* = 0.36; (**b**) *ε* = 0.69; (**c**) *ε* = 1.20; (**d**) the fraction of different misorientation angle.

**Figure 12 materials-17-01697-f012:**
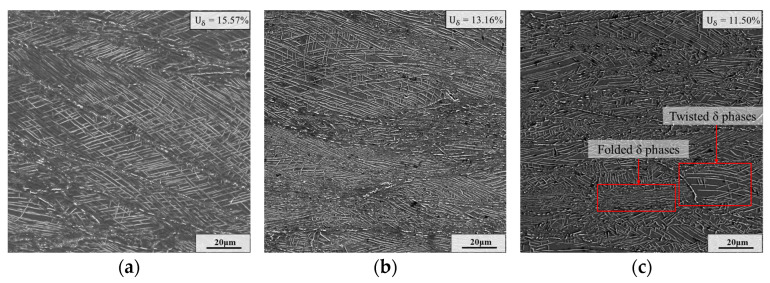
SEM micrographs of samples II after final deformation with different final strains: (**a**) *ε* = 0.36; (**b**) *ε* = 0.69; (**c**) *ε* = 1.20.

**Figure 13 materials-17-01697-f013:**
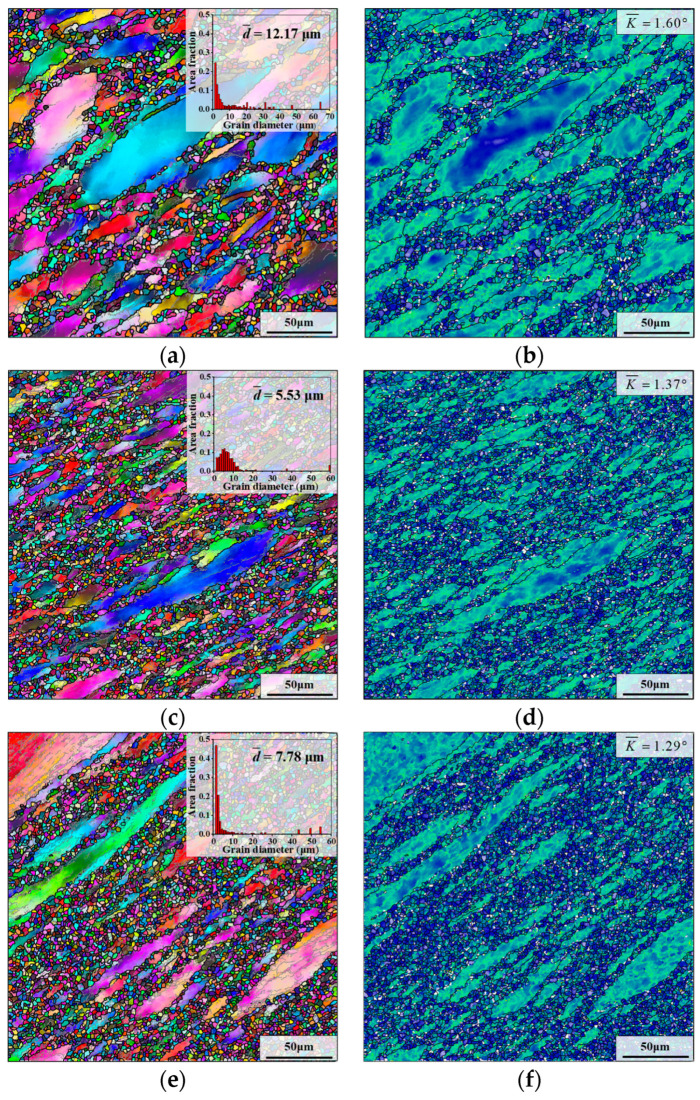
Deformed grain microstructures of samples II with different holding times: (**a**,**b**) *t*_hold_ = 15 min; (**c**,**d**) *t*_hold_ = 30 min; (**e**,**f**) *t*_hold_ = 60 min. (The holding temperature is 980 °C and deformation parameters are *T* = 980 °C, $\dot{\varepsilon }=0.1$ s^−1^, *ε* = 0.69).

**Figure 14 materials-17-01697-f014:**
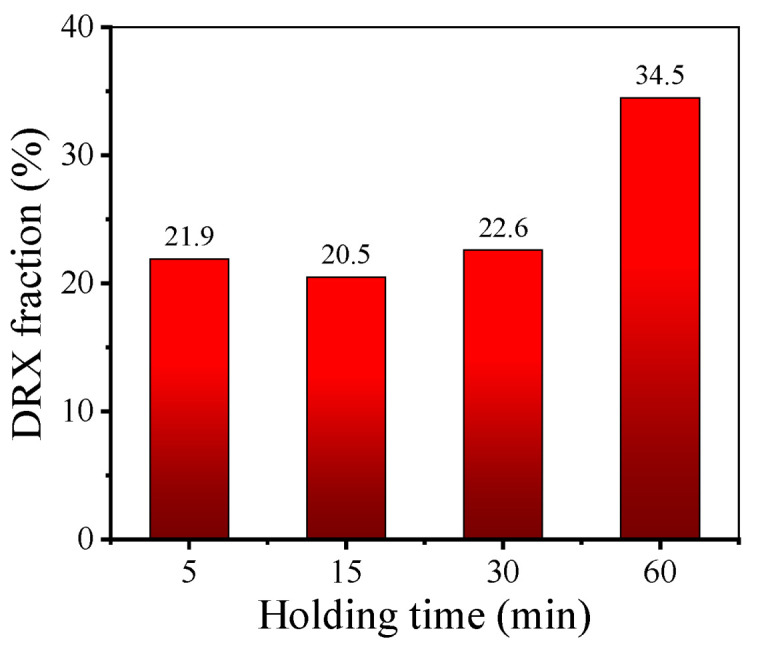
DRX fraction of deformed grain microstructures of samples II with different holding times.

**Figure 15 materials-17-01697-f015:**
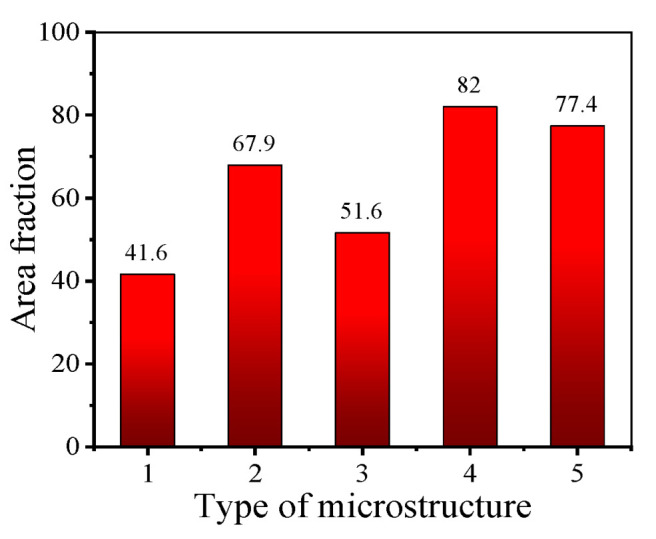
Area frequency of new-born grains in the microstructures after final deformation.

**Figure 16 materials-17-01697-f016:**
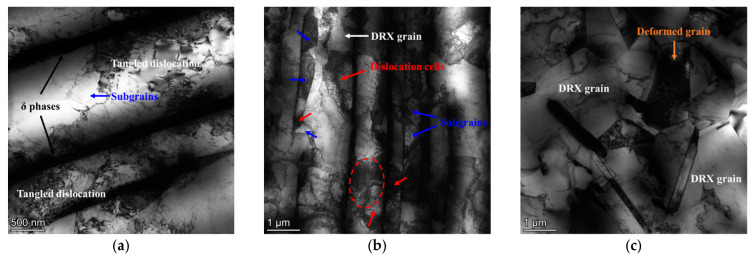
Bright-field TEM images of samples II deformed at 980 °C–0.1 s^−1^ to different final strains: (**a**) *ε* = 0.36; (**b**) *ε* = 0.69; (**c**) *ε* = 1.20. (The red circle indicates the meeting of the dislocation cells).

**Figure 17 materials-17-01697-f017:**
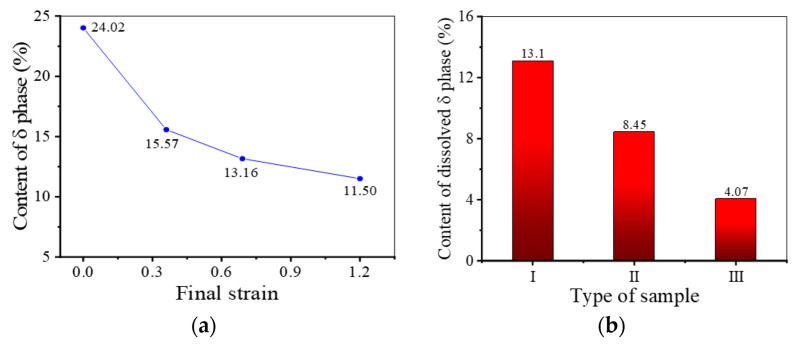
Diagrams of: (**a**) δ phase content of samples II after final deformation with different final strains at 980 °C and 0.1 s^−1^; (**b**) content of dissolved δ phase in sample I–III during final deformation (980 °C–0.1 s^−1^–0.69).

**Figure 18 materials-17-01697-f018:**
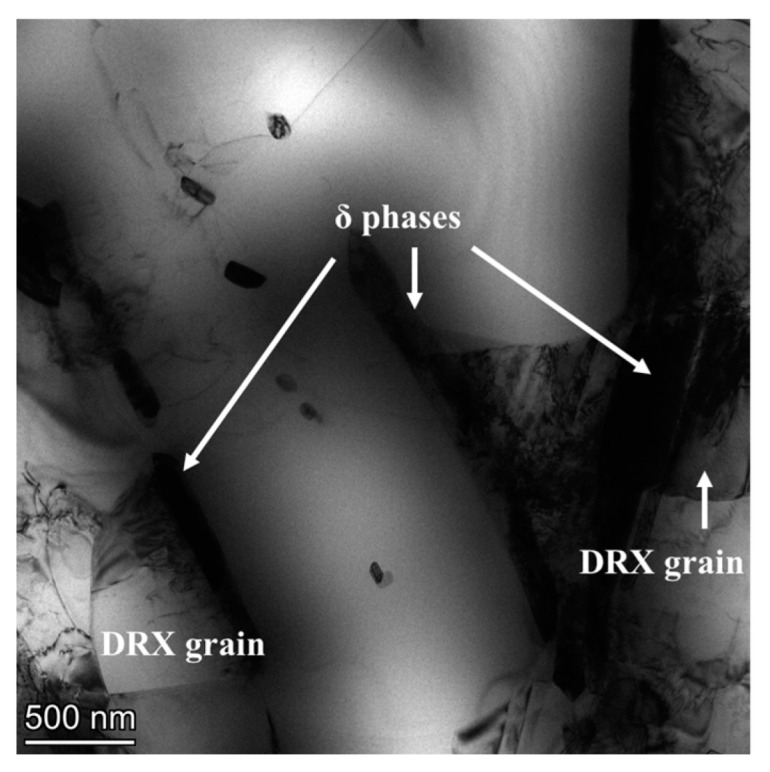
The bright-field TEM image of samples III deformed at 980 °C–0.1 s^−1^ to a final strain of 0.69.

**Table 1 materials-17-01697-t001:** Chemical composition (wt %) of the studied GH4169 superalloy.

Element	Ni	Cr	Nb	Mo	Ti	Al	Co	C	Fe
wt %	52.82	18.96	5.23	3.01	1.00	0.59	0.01	0.03	bal.

**Table 2 materials-17-01697-t002:** The typical grain microstructure types after HTH and corresponding processes.

Type	Grain Microstructure Characteristics	Corresponding Process
1	$\mathrm{Coarse}\;\mathrm{and}\;\mathrm{uniform},\;\bar{K}=0.88{}^{\circ}$, $\bar{d}=41.44\;\mu m$	Sample I held at 980 °C for 5 min
2	$\mathrm{Mixed},\;\bar{K}=1.25{}^{\circ}$, $\bar{d}=29.09\;\mu m$	Sample II held at 980 °C for 5 min
3	$\mathrm{Mixed},\;\bar{K}=1.10{}^{\circ}$, $\bar{d}=22.66\;\mu m$	Sample II held at 980 °C for 15 min
4	$\mathrm{Mixed},\;\bar{K}=1.24{}^{\circ}$, $\bar{d}=20.62\;\mu m$	Sample III held at 980 °C for 5 min
5	$\mathrm{Fine},\;\bar{K}=0.61{}^{\circ}$, $\;\bar{d}=15.92\;\mu m$	Sample II held at 980 °C for 60 min

## Data Availability

The raw/processed data required to reproduce these findings cannot be shared at this time as the data also form part of an ongoing study.

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
