# Peer review of "Investigation on Mechanism of Microstructure Evolution during Multi-Process Hot Forming of GH4169 Superalloy Forging"

_materials, 2024, doi:10.3390/ma17071697_

Round 1

Reviewer 1 Report

Comments and Suggestions for Authors

In general, the manuscript is well-structured, however:

It is necessary and very important to include by means of scientific references and theoretical reasons:

What are the possible technological applications?

What are the possible researches in the future? 

Comments on the Quality of English Language

Some passive voices can improve

 Further prolonging the time, the behavior of static recrystallization nucleation is weakened due to the dissolution of δ phase content and the consumption of deformation energy, which can be confirmed by the trend of the number of newborn grains.

Further prolonging the time, the behavior of static recrystallization nucleation is weakened due to the dissolution of δ phase content and the consumption of deformation energy, which the trend of the number of newborn grains can confirm.

More simplicity to improve clarity:

it can be seen that many low-angle grain boundaries

many low-angle grain boundaries

The reason why the average grain size of sample I is the largest is that the content of small recrystallized grains in its microstructure is the least

The average grain size of sample I is the largest because the content of small recrystallized grains  in its microstructure is the least

Microstructure of the sample  

Sample microstructure

Grammatics changes are necessary

After HTH, there are three typical grain microstructures, namely uniform coarse grain microstructure, mixed grain microstructure and fine grain microstructure.

After HTH, there are three typical grain microstructures: uniform coarse grain microstructure, mixed grain microstructure, and fine grain microstructure.

Author Response

Dear Mr. Mihai Galeteanu and Reviewers:

Thanks for the suggestive comments from reviewers and editor. These comments are very valuable and helpful for the authors to improve the paper. The manuscript has been carefully checked again. The following are the response to the comments from the reviewer and editor. The corresponding modifications have been made and the important corrections are highlighted in the YELLOW color in the revised manuscript.

Author's Reply to the Review Report (Reviewer 1):

In general, the manuscript is well-structured, however: It is necessary and very important to include by means of scientific references and theoretical reasons:

  1. Some passive voices can improve:

Further prolonging the time, the behavior of static recrystallization nucleation is weakened due to the dissolution of δ phase content and the consumption of deformation energy, which can be confirmed by the trend of the number of newborn grains.

Response: Thanks for the reviewer’s constructive comments. According to the constructive comments, the corresponding sentence has been adjusted. The detailed modifications are listed as follows.

Section “3.3.2. Effect of holding time”

“……Further prolonging the time, the behavior of static recrystallization nucleation is weakened due to the dissolution of δ phase and the consumption of deformation energy. The trend of the number of newborn grains can confirm this idea……”

  1. More simplicity to improve clarity:

it can be seen that many low-angle grain boundariesmany

The reason why the average grain size of sample I is the largest is that the content of small recrystallized grains in its microstructure is the least.

Microstructure of the sample

Response: Thanks for the reviewer’s constructive comments. We have simplified the sentences pointed out by the reviewer. The corresponding modifications have been made in the revised manuscript. Thanks!

  1. Grammatics changes are necessary

After HTH, there are three typical grain microstructures, namely uniform coarse grain microstructure, mixed grain microstructure and fine grain microstructure.

Response: Thanks for the reviewer’s constructive comments. According to the constructive comments, the corresponding sentence has been adjusted. The detailed modifications are listed as follows.

Section “4. Discussion”

“……After HTH, there are three typical grain microstructures: uniform coarse grain micro-structure, mixed grain microstructure, and fine grain microstructure……”

Reviewer 2 Report

Comments and Suggestions for Authors

Formal remarks

- mind capitalization in the legend of the figures

- use numbers for citations

- mind font type and font size

Materials and experiments

- parameters of solution treatment should be written in words

- "where the temperature and the strain rate of
pre-deformation are 950 °C and 0.1 s-1, respectively" sounds misunderstanding, should be explained clearly

- how were the samples cut? Can cutting influence the measured properties?

- Fig.3: Figure legend text should state for which experiment this experimental procedure was applied

- Should be explained, how twins are recognized (for example in the text before Fig.1

Comments on the Quality of English Language

Minor editing is sufficient. Especially the formal errors should be addressed.

Author Response

Dear Mr. Mihai Galeteanu and Reviewers:

Thanks for the suggestive comments from reviewers and editor. These comments are very valuable and helpful for the authors to improve the paper. The manuscript has been carefully checked again. The following are the response to the comments from the reviewer and editor. The corresponding modifications have been made and the important corrections are highlighted in the YELLOW color in the revised manuscript.

Author's Reply to the Review Report (Reviewer 2):

  1. Formal remarks

- mind capitalization in the legend of the figures

- use numbers for citations

- mind font type and font size

Response: Thanks for the reviewer’s constructive comments. The corresponding modifications have been made in the revised manuscript. Thanks!

Materials and experiments

  1. parameters of solution treatment should be written in words and "where the temperature and the strain rate of pre-deformation are 950 °C and 0.1 s-1, respectively" sounds misunderstanding, should be explained clearly.

Response: Thanks for the reviewer’s constructive comments. According to the constructive comments, parameters of solution treatment have been re-written in words and the parameters of hot-upsetting have been redescribed. The detailed modifications are listed as follows.

Section “2. Materials and experiments”

“……Then, the experimental billet was subjected to solution treatment. Solution treatment is that the billet is held at 1040 °C for 85 min……Subsequently, hot-upsetting (i.e., pre-deformation) was applied to the experimental billet, reducing its height to 70 mm. The temperature of hot-upsetting is 950 °C and the strain rate is 0.1 s-1……”

  1. How were the samples cut? Can cutting influence the measured properties?

Response: Thanks for the reviewer’s constructive comments. All experimental samples were cut by CNC wire cutting machines. The process of cutting the sample has minimal impact on the measured properties. On the one hand, superalloys have excellent microstructural stability. On the other hand, the surfaces of all samples were polished to remove oxide layers and cutting marks. Thanks.

  1. Fig.3: Figure legend text should state for which experiment this experimental procedure was applied

Response: Thanks for the reviewer’s constructive comments. According to the constructive comments, we have added a description in the legend text in Fig. 3 to state for which experiment this experimental procedure was applied. In addition, corresponding descriptions have been added in the text.

  1. Should be explained, how twins are recognized (for example in the text before Fig.1)

Response: Thanks for the reviewer’s constructive comments. According to the constructive comments, we have been added a description for characteristics of annealing twins in the text before Fig. 1. The detailed modifications are listed as follows.

Section “2. Materials and experiments”

“……Typically, annealing twins are observed within equiaxed grains and have mutually par-allel grain boundaries. As displayed in Fig. 1, the grain microstructure of GH4169 super-alloy after solution treatment consists of equiaxed grains and annealing twins……”

Reviewer 3 Report

Comments and Suggestions for Authors

In the manuscript titled "Investigation on the Mechanism of Microstructure Evolution During Multi-Process Hot Forming of GH4169 Superalloy Forging," the authors present a variety of findings concerning the microstructure evolution of GH4169 superalloy. Despite the clear presentation of results, several issues must be addressed before recommending the manuscript for acceptance in the MDPI Materials journal:

1. In the introduction, it is recommended to provide information on the typical elemental composition of GH4169 to acquaint the reader with the alloy's complexity and the significance of the research problem.

2. In Section 2, it is crucial to list all commercially sourced materials and reagents, including the details of the commercial provider for the GH4169 alloy.

3. Section 2 should also include the technical specifications of the equipment used to characterize the samples, as well as the parameters used in each characterization technique, to ensure the experiments' reproducibility.

4. The figure captions currently located below the images should be integrated into the general figure description to prevent confusion among readers.

5. The type of detector used to capture the SEM images is not specified. It appears a secondary electrons detector was used. Including images from both secondary electron and backscattered electron detectors would enrich the discussion.

6. All figures must be referenced in the manuscript text. For instance, Figures 5a, b, c, and d are not mentioned in the main body of the text.

7. Regarding Figure 16, details on the sample preparation for TEM analysis should be provided.

8. Prior to the conclusion section, a discussion on the potential impact of temperature on the elongated, needle-like δ phase, in comparison to other research findings, is advisable. Specifically, the influence of oxidation on microstructural changes should be considered. I recommend citing and discussing this topic in relation to the following article and related research on thermal treatment or thermal oxidation:

H. Rojas-Chávez, et al., "The formation of ZnO structures using thermal oxidation: How a previous chemical etching favors either needle-like or cross-linked structures," Materials Science in Semiconductor Processing, Volume 108, 2020, 104888, ISSN 1369-8001, https://doi.org/10.1016/j.mssp.2019.104888

9. Lastly, it is suggested to add a paragraph near the conclusion section that highlights the innovation of this research compared to other similar studies.

Author Response

Dear Mr. Mihai Galeteanu and Reviewers:

Thanks for the suggestive comments from reviewers and editor. These comments are very valuable and helpful for the authors to improve the paper. The manuscript has been carefully checked again. The following are the response to the comments from the reviewer and editor. The corresponding modifications have been made and the important corrections are highlighted in the YELLOW color in the revised manuscript.

Author's Reply to the Review Report (Reviewer 3):

In the manuscript titled "Investigation on the Mechanism of Microstructure Evolution During Multi-Process Hot Forming of GH4169 Superalloy Forging," the authors present a variety of findings concerning the microstructure evolution of GH4169 superalloy. Despite the clear presentation of results, several issues must be addressed before recommending the manuscript for acceptance in the MDPI Materials journal:

  1. In the introduction, it is recommended to provide information on the typical elemental composition of GH4169 to acquaint the reader with the alloy's complexity and the significance of the research problem.

Response: Thanks for the reviewer’s constructive comments. According to the constructive comments, the information on the typical elemental composition of GH4169 has been provided in the introduction. The detailed modifications are listed as follows.

Section “1. Introdution”

“GH4169 alloy is a typical Ni-Fe-Cr based superalloy, which can maintain excellent comprehensive mechanical properties and microstructure stability under high temperatures and high pressure……”

  1. In Section 2, it is crucial to list all commercially sourced materials and reagents, including the details of the commercial provider for the GH4169 alloy.

Response: Thanks for the reviewer’s constructive comments. Table 1 shows the detailed chemical composition of GH4169 superalloy. In addition, according to the constructive comments, the information on suppliers of GH4169 alloy we used has been added in Section Materials and experiments. The detailed modifications are listed as follows.

Section “2. Materials and experiments”

“Table 1 lists the detailed chemical compositions of a commercial GH4169 superalloy that is provided by Fushun Special Steel Co., Ltd. in China……”

  1. Section 2 should also include the technical specifications of the equipment used to characterize the samples, as well as the parameters used in each characterization technique, to ensure the experiments' reproducibility.

Response: Thanks for the reviewer’s constructive comments. According to the constructive comments, we have added the technical specifications of the equipment used to characterize the samples and the parameters used in each characterization technique in Section 2. The detailed modifications are listed as follows.

Section “2. Materials and experiments”

“……In this study, to obtain the microstructures of samples, we used optical microscopy (OM), Scanning Electron Microscopy (SEM), Electron Backscattered Diffraction (EBSD), and Transmission Electron Microscopy (TEM). For SEM and OM, the samples were firstly mirror polished, and then the samples were electrolytically corroded in the etching solution composed of 70 ml H3PO4 and 30 mm H2O. The voltage of electrolytic corrosion was 3 V and the electrolytic corrosion time was about 10-25 s. In the TESCAN MIRA4 LMH Scanning Electron Microscope, several images with magnifications of 1000x, 2000x and 5000x were acquired using secondary electron signals. The model of OM device is Keyence VHX-5000. For EBSD and TEM, discs with a diameter of 3 mm were cut from smooth flakes with a thickness of 70-80 mm through a hole punch, and then the discs were electrolytically polished by a solution of 10 mL HClO4 and 90 mL CH3CH2OH. The temperature of electrolytic polishing was -40 to -25 °C the voltage was 25 V. The model of EBSD equipment is NordlysMax2. In EBSD characterization, the observation area was a square of 250 μm × 250 μm and the scanning step was set to 1 μm. Besides, all EBSD data were performed on the MTEX-5.7.0 toolbox. When reconstructing grains, low-angle grain boundaries (LAGBs, 2°≤ θ < 15°) are repre-sented by gray lines and high-angle grain boundaries (HAGBs, θ > 15°) are represented by black lines. The model of TEM device is Talos FEI 200X.”

  1. The figure captions currently located below the images should be integrated into the general figure description to prevent confusion among readers.

Response: Thanks for the reviewer’s constructive comments. According to the constructive comments, all text descriptions located below the images have been integrated into the general figure descriptions. The corresponding modifications have been made in the revised manuscript. Thanks!

  1. The type of detector used to capture the SEM images is not specified. It appears a secondary electrons detector was used. Including images from both secondary electron and backscattered electron detectors would enrich the discussion.

Response: Thanks for the reviewer’s constructive comments. Indeed, the type of detector we used to capture the SEM images was the secondary electrons detector. The corresponding modifications have been made in the revised manuscript. Thanks!

  1. All figures must be referenced in the manuscript text. For instance, Figures 5a, b, c, and d are not mentioned in the main body of the text.

Response: Thanks for the reviewer’s constructive comments. According to the constructive comments, we have added descriptions for some figures in the main text of the manuscript to make all figures be referenced. The corresponding modifications have been made in the revised manuscript. Thanks!

  1. Regarding Figure 16, details on the sample preparation for TEM analysis should be provided.

Response: Thanks for the reviewer’s constructive comments. According to the constructive comments, the details on the sample preparation for TEM analysis have been added in Section 2. The detailed modifications are listed as follows.

Section “2. Materials and experiments”

“……For EBSD and TEM, discs with a diameter of 3 mm were cut from smooth flakes with a thickness of 70-80 mm through a hole punch, and then the discs were electrolytically polished by a solution of 10 mL HClO4 and 90 mL CH3CH2OH. The temperature of electrolytic polishing was -40 to -25 °C the voltage was 25 V……”

  1. Prior to the conclusion section, a discussion on the potential impact of temperature on the elongated, needle-like δ phase, in comparison to other research findings, is advisable. Specifically, the influence of oxidation on microstructural changes should be considered. I recommend citing and discussing this topic in relation to the following article and related research on thermal treatment or thermal oxidation:
  2. Rojas-Chávez, et al., "The formation of ZnO structures using thermal oxidation: How a previous chemical etching favors either needle-like or cross-linked structures," Materials Science in Semiconductor Processing, Volume 108, 2020, 104888, ISSN 1369-8001, https://doi.org/10.1016/j.mssp.2019.104888

Response: Thanks for the reviewer’s constructive comments. Both temperature and δ phase have significant effects on the final microstructure. Indeed, it is interesting and necessary to discuss the effect of temperature on the long needle-like δ phase. The relevant discussion has been added in the Section 4.2. The detailed modifications are listed as follows.

Section “4.2 Interaction mechanisms between δ phase and DRX nucleation”

“Furthermore, compared with the short rod-like or the spherical δ phase, the long nee-dle-like δ phase has a stronger promotion effect on the DRX behavior. Increasing the final deformation although reduce the critical nucleation dislocation density, eliminates part of long needle-like δ phases. Therefore, coordinating the deformation temperature and the δ phase morphology is the key to obtaining a uniform and fine deformation grain micro-structure [1,2].

[1] Singh, V.K.; Sahoo, D.; Amirthalingam, M.; Karagadde, S.; Mishra, S.K. Dissolution of the Laves Phase and δ-Precipitate Formation Mechanism in Additively Manufactured Inconel 718 During Post Printing Heat Treatments. Addit. Manuf. 2024, 81, 104021.

[2] Rojas-Chávez, H.; Cruz-Martínez, H.; Montejo-Alvaro, F.; Farías, R.; Hernández-Rodríguez, Y.M.; Guillen-Cervantes, A.; Ávila-García, A.; Cayetano-Castro, N.; Medina, D.I.; Cigarroa-Mayorga, O.E. The Formation of Zno Structures Using Thermal Oxidation: How a Previous Chemical Etching Favors Either Needle-Like Or Cross-Linked Structures. Mater. Sci. Semicond. Process. 2020, 108, 104888.

  1. Lastly, it is suggested to add a paragraph near the conclusion section that highlights the innovation of this research compared to other similar studies.

Response: Thanks for the reviewer’s constructive comments. According to the constructive comments, we have been supplemented a paragraph that highlights the innovation of this research compared to other similar studies. The detailed modifications are listed as follows.

Section “5. Conclusions”

“Different from existing studies that only analyze the microstructure evolution of GH4169 alloy during a single hot deformation, this study focuses on the microstructure evolution during multi-process hot forming of GH4169 superalloy and the relationship between the microstructure evolution in each process……”

Author's Reply to the Editor:

  1. Upon careful review of your submitted papers, we have observed that the proportion of references from Chinese authors exceeds one-third of the total citations. As per our guidelines, we kindly request that you consider revising this distribution to ensure adherence to the stipulated maximum limit, where references from a single country should not surpass one-third of the total citations. We also observed the self citation rate of your manuscript is over our admitted maximum. We kindly suggest to the authors to replace some of the self-cited references with some recent related ones from other research groups to reduce the self-citation rate to less than 15%. We believe this will increase the state of the art of this paper. If the self-citation cannot be reduced within 15%, please provide the academic reason.

Response: Thanks for the editor’s constructive comments. According to the constructive comments, we have revised the references. The corresponding modifications have been made in the revised manuscript. Thanks!
